# Model of Groundwater Flow Using Boltzmann Lattice-Gas Automation Method In Maros Karst Region, Indonesia

Muhammad Arsyad<sup>1</sup>, Nasrul Ihsan<sup>1</sup>, Vistarani Arini Tiwow<sup>1</sup>, Ansari Saleh Ahmar<sup>2</sup>

<sup>1</sup>Department of Physics, Universitas Negeri Makassar, Makassar, 90223, Indonesia <sup>2</sup> Department of Statistics, Universitas Negeri Makassar, Makassar, 90223, Indonesia

*Correspondence to*: Muhammad Arsyad (m arsyad288@unm.ac.id)

**Abstract.** In this study, modeling of mineral resources of underground water in karst mountain area of Maros-Pangkep, Indonesia was made by the Boltzmann lattice-gas automata method. This method was applied to solve the problem of groundwater flow by viewing them as a lattice gas. Simulation models of groundwater flow in the form of 16 and 32 particle

size with a barrier particles form a circle on the plate and different variations of time to reach steady its state. The simulation results showed that the greater the time duration, the more different fluid motion pattern. For the state of the karst region, this means that the heterogeneous medium karst very unstable and affect the movement of the particles. In addition, karstification process takes place in a relatively long time with adequate rainfall.

#### **1** Introduction

Maros karst region has a water system that is conducive to the life of the surrounding population. Availability of groundwater in karst areas with good potential water sources in the area ranged from 4.03 to 45.1 ohm-meter (Arsyad, 2002). Rock lithology is composed of a material not solid, compact and solid to hard.

Karst region gained serious attention because of its ability to store water in the form of rivers ground under the cave. The result will directly influence the physical properties of the medium include porosity, density, permeability and permitivity of

20 the medium (Arsyad, 2009). The physical properties of the medium was obtained by testing rock samples to determine the physical quantities of ground water in the form of medium porosity ( $\phi$ ), soil particle density ( $\rho$ ), and the medium permeability ( $\kappa$ ) (Arsyad, 2010a).

The results showed that the structure of the medium in the karst region including porous media (Arsyad, 2010b). Fluid flow in porous media with restrictions that are quite complex, so in the modeling required special treatment as well. To that end,

in fluid mechanics introduced their discrete lattice gas called lattice-gas automaton (Hughes et al., 1999). System fluid flow dynamics can be described as a collection of a large number of molecules that interact according to Newton's equation of motion (Khotimah and Liong, 2002).

Lattice gas automata method can be used for several different types of mixed fluid flow characteristics or fluid through the medium of complex geometry structure (Arsyad, 1999; Supriyatno, 1998). Model-lattice Boltzmann gas consists of particles that move from one cell to another in a triangular lattice (Rothman, 1990).

Simulations will be made showed that the hydrodynamic flow pattern with particles or molecular dynamics simulations
(Aharonof and Rothman, 1993; Rothman, 1988). Such modeling was applied to solve fluid flow problems by viewing them as lattice-gas (Khotimah and Liong, 2002), especially for heterogeneous medium with complex boundaries, such as medium karst region.

## 2 Research Method

Location of sampling research conducted in Maros, located between 40°45'50" LS and 109°20'00" lon.E up with 129°12'00"

Ion.E. Furthermore, made simulations to model groundwater flow patterns using computer of 2930 Intel Aspire computer, Intel<sup>®</sup> Core <sup>TM</sup> Duo processor T6400 (2.0 Ghz, 800 MHz FSB, 2 MB L2 cache), Mobile Intel<sup>®</sup> Graphics Media Accelerator 4500 MHD, 1 GB DDR2, 12.1" WXGA Acer CrystalBrite<sup>TM</sup> LCD, 250 GB HDD, DVD-Super Multi DL, dan 802.11 a/b/g/draft-N WLAN.

The data required to perform the simulation model of groundwater flow in karst region are the physical properties of the

15 medium Maros karst region that includes, coefficient of permeability, porosity and density of the medium as shown in Table 1.

This study uses the time t and t + n as the data to simulate fluid flow, so that:

- 1. Data on the physical properties of the medium are constant, the porosity ( $\phi$ ), soil particle density ( $\rho$ ), and the permeability of the medium ( $\kappa$ ) as in the table 1.
- 2. Simulation with Lattice-Gas is the model used to facilitate molecular movement known as the Lattice-Gas Automata. Model Lattice-Gas Boltzmann consists of particles that move from one cell to another in a triangular lattice. Using the lattice equilateral triangle of each particle can have six possible directions speed. Each particle have a mass unit and one unit of speed. The total momentum of the collision is zero and the particle rotated 60 degrees. Because the particles in the Gas-Lattice Boltzmann be described as a boolean variable  $n_i(x, t)$ , with the direction determined by the speed unit
- (Wolf-Gladrow, 2000):

$$c_i = \left[\cos\left(\frac{2i}{6}\right), \sin\left(\frac{2i}{6}\right)\right] \quad \text{with } i = 1, 2, \dots, 6$$
 (1)

so that the particle moves by following a mathematical equation and is written:

$$n_i(x+c_i,t+1) = n_i(x,t) + \Delta_i(n_i(x,t))$$
(2)

 $n_i(x+c_i,t+1)$  is a boolean. If there are particles then it value is 1, and 0 if no moving particles from the collision position 30 x to position  $x + c_i$  at time t + 1.

For notation  $n_i(x,t)$  expressed particle moving to the collision position x of the direction  $c_i$  at time t. As for  $\Delta_i(n_i(x,t))$  is the collision operator that describes changes in the value  $n_i(x, t)$ . These collisions can be valuable operator 0, +1, or -1. If there is no change in the number of particles in the direction *i* caused by collision events, is the number of particles before and after the collision in the direction *i* are same, then the value of  $n_i(x,t) + \Delta_i(n_i(x,t))$  is 0.

| Dimonsions                           | TI                 | Soil Types |          |
|--------------------------------------|--------------------|------------|----------|
| Dimensions                           | Unit               | Clay       | Chalk    |
| Spesific Grafity (Gs)                | -                  | 2.80       | 2.61     |
| Wet Density, $\rho_{wet} = (4)/(5)$  | Gr/Cm <sup>3</sup> | 1.05       | 1.74     |
| Porosity ( $\phi$ )                  | %                  | 77.30      | 45.17    |
|                                      |                    | (High)     | (High)   |
| Permeability Coeficient ( $\kappa$ ) | cm/det             | 8.14E-03   | 7.33E-08 |
| (Constant Head Permeameter)          |                    | (Low)      | (Low)    |
| Permeability Coeficient ( $\kappa$ ) | cm/det             | 1.35E-03   | 7.33E-08 |
| (Falling Head Permeameter)           |                    | (Low)      | (Low)    |
| Degree of Saturation (Sr)            | %                  | 53.48      | 68.16    |

Table 1. Physical Medium Price of Maros karst region

The price in Table 1, further processed to obtain the movement of ground water medium in Maros karst region, the algorithm as follows

as follows

- 1) Input parameters of time duration flow by N units of time.
- 2) create a medium with properties of the medium as in Table 1.
  - 3) placing the initial position of particles in a triangular lattice such that the total momentum of the collision is zero and the particles allows rotated 60 degrees (Rothman, 1988).
  - Particles in the Boltzmann lattice gas can be described as a boolean variable n<sub>i</sub> (x,t), the direction is determined by the speed unit of

| <i>c</i> <sub><i>i</i></sub> = | $\left[\cos\left(\frac{2i}{6}\right),\right]$ | sin | $\left(\frac{2i}{6}\right)$ |
|--------------------------------|-----------------------------------------------|-----|-----------------------------|
|--------------------------------|-----------------------------------------------|-----|-----------------------------|

with i = 1, 2, ..., 6.

- 5) Particle motion, every one unit of time, the particles move in the direction of the lattice unit speed.
- 6) Tracking was done at each lattice configuration to determine whether there was the possibility of change in the direction of the particles caused by the collision with a wall or occurs whether or not the configuration, so that : a) if there was a
- change in the direction of the particle or configuration changes, then the transformation was done on the basis of the data, and b ) if no change in the direction of the particle or configuration changes, then the program straight to step 6
  - 7) Repeat steps (1) to (5) was performed until the time parameter achieve N as unit time given input.
  - 8) Display the results

9) Finish.

The simulation began with the movement of fluid due to the difference in pressure on both the left and right sides. Fluid moves until it reaches equilibrium. Equilibrium occurs if the time is still moving, but the flow has shown a similar pattern. That is, the time is growing but the flow pattern remains the same. So the simulation stopped if the flow pattern does not change anymore.

In this study, it doing extraction/pumping out of water. The movement of groundwater in medium karst areas is very different with medium nonkarst. The heterogeneous of medium characteristic effect on fluid motion. Continuously of fluid motion obtain changes of pressure gradient will cause the process of kartisification be slow. Karstification will determine the characteristic of constantly changing of medium in proportion to time. If the pattern of flow shown in this simulation is not

- changed, then karstification started, thus permeable of medium karst becomes impermeable, so that the medium be a water reservoir that holds water in a relatively long time. This process will lead to the karst region has uvala, polje and karst characteristics as water tendon is different from other nonkarst medium. And Selection of a barrier in the form of plate adapted to the characteristic of medium karst is vertical barrier wall. Dimensions of plate depending on the size of the Lattice Boltzmann that allows particles to move with 6 possible directions as equation (1).
- For barriers of circle adapted to the characteristic of medium karst that has undergone karstification. Karstification occur if the medium that was rigid and stiff, because dissolution occurs deformation. This deformation caused barrier fluid movement turns into a circle particles with dimensions larger than the dimensions of the particles that started from 16, 32, etc.

#### **3 Result and Discussion**

Simulation models of groundwater flow in Maros karst region were divided into two major parts of the groundwater flow model of 16 particle size with a barrier in the form of plates and circles, as well as ground water flow model of 32 particle size with a barrier in the form of plates and circle, at a different time variation to reach steady state. The results are shown in Figure 1 and Figure 2.

30