# Peer review of "Model of Groundwater Flow Using Boltzmann Lattice-Gas Automation Method In Maros Karst Region, Indonesia"

_Drinking Water Engineering and Science, 2016_

## Short Comment (SC1) · 11 May 2017

Dear author

I have read your paper, I see you choose Maros karst region. Why you choose this location? And You using Boltzmann Lattice-Gas Automation Method, why you not using other method, for example the Lattice Gas Cellular Automata model of Frisch, Hasslacher and Pomeau? What the advantages this method?

---

## Short Comment (SC2) · 25 May 2017

We choose Maros because it is one of karst region in Indonesia.

The problems of Lattice Gas Cellular Automata were mostly resolved if we using the Lattice Boltzmann Models.

In Lattice Boltzmann Models, the velocities $c_i$ of the fluid particles are discrete, but we average over an ensemble of systems to obtain a probabilistic distribution $n_i$ of velocities.

The advantage of Lattice Boltzmann Models: the fluid density and velocity in Lattice Boltzmann Models are both continuous, which makes it possible to satisfy the invariance of Galilean and moreover to avoid other problems with the Lattice Gas Model.

You can others advantage of these models in: http://www.leb.eei.uni-erlangen.de/winterakademie/2011/report/content/course02/pdf/0208.pdf and you can read the book with the title "Lattice Gas Methods: Theory, Applications, and Hardware" by Gary D. Doolen.

---

## Referee Comment (RC1) · Anonymous Referee #1 · 26 Jun 2017

I have to check your paper: a. In line 10-11 page 5, In figure 1, you must be consistency. There is word "unit of time" and "second". What are you using? "unit of time" or "second"? b. You must corrections the citation, in line 14 page 6.

---

## Referee Comment (RC2) · M. Choudhary (Referee) · 26 Jun 2017

The authors have proposed a model for the groundwater flow in the Karst Region. They have used "Boltzmann Lattice-Gas Automation Method" In Maros Karst Region, Indonesia, and shown that greater the time duration, there will be more different fluid flow patterns. The authors claim to model the mineral resources in Karst Mountain but do not provide any data pertaining to mineral resources of the region. The manuscript has spelling and grammatical errors. The authors are advised to revise the paper so that the logical flow is maintained. Some of the references in the list are not cited in the text. Overall write up of the paper should be improved.

---

## Author Comment (AC1) · 17 Jul 2017

Thanks for your review. I inform that we using unit of time (second).

---

## Author Comment (AC2) · 17 Jul 2017

Thanks for review

The authors claim to model the mineral resources in Karst Mountain but do not provide any data pertaining to mineral resources of the region. - I have added in submission

The manuscript has spelling and grammatical errors. - I have revised

Overall, I have improved the write up of the paper.

Please also note the supplement to this comment:

[Figure]

https://www.drink-water-eng-sci-discuss.net/dwes-2016-9/dwes-2016-9-AC2-supplement.pdf

[Figure]

**Supplement:**

**Model of Groundwater Flow Using Boltzmann Lattice-Gas Automation Method In Maros Karst Region, Indonesia**

Muhammad Arsyad[1], Nasrul Ihsan[1], Vistarani Arini Tiwow[1], Ansari Saleh Ahmar[2]

[1]Department of Physics, Universitas Negeri Makassar, Makassar, 90223, Indonesia
[2] Department of Statistics, Universitas Negeri Makassar, Makassar, 90223, Indonesia

*Correspondence to*: Muhammad Arsyad (m_arsyad288@unm.ac.id)

**Abstract.** In this study, modelling of mineral resources of underground water in Karst mountain area of Maros-Pangkep, Indonesia was created by the Boltzmann lattice-gas automata method. This method was applied to solve the problem of groundwater flow by viewing them as a lattice gas. Simulation models of groundwater flow in the form of 16 and 32 particle sizes with barrier particles form a circle on the plate and different variations of time to reach a steady state. The simulation results showed that the greater the time duration, the more different fluid motion pattern. For the state of the Karst region, this means that the heterogeneous medium of Karst is very unstable and affects the movement of the particles. In addition, karstification process takes place in a relatively long time with adequate rainfall.

**1 Introduction**

Maros Karst region has a water system that is conducive to the life of the surrounding population. Availability of the groundwater in Karst areas with good potential water sources in the area ranged from 4.03 to 45.1 ohm-meter (Arsyad, 2002). Rock lithology is composed from a soft, compact and hard material.

Karst region gained serious attention because of its ability to store water in the form of rivers ground under the cave. The result will directly influence the physical properties of the medium including porosity, density, permeability and permittivity of the medium (Arsyad, 2009). The physical properties of the medium was obtained by testing rock samples to determine the physical quantities of ground water in the form of medium porosity ($\phi$), soil particle density ($\rho$), and the medium permeability ($\kappa$) (Arsyad, 2010a).

The results showed that the structure of the medium in the Karst region including porous media (Arsyad, 2010b). Fluid flow in porous media with restrictions that are quite complex, so the modelling requires special treatment as well. To that end, in fluid mechanics introduced their discrete lattice gas called lattice-gas automaton (Hughes et al., 1999). System fluid flow dynamics can be described as a collection of a large number of molecules that interact according to Newton's equation of motion (Khotimah and Liong, 2002).

Lattice gas automata method can be used for several different types of mixed fluid flow characteristics or fluid through the medium of complex geometry structure (Arsyad, 1999; Supriyatno, 1998). Model-lattice Boltzmann gas consists of particles that move from one cell to another in a triangular lattice (Rothman, 1990).

The upcoming simulations showed that the hydrodynamic flow pattern are similar to either particles or molecular dynamics simulations (Aharonof and Rothman, 1993; Rothman, 1988). Such modelling was applied to solve fluid flow problems by viewing them as lattice-gas (Khotimah and Liong, 2002), especially for heterogeneous medium with complex boundaries, such as Karst region medium.

**2 Research Method**

Location of sampling research conducted in Maros was located between 40°45'50" LS and 109°20'00" lon.E up with 129°12'00" lon.E. Furthermore, simulations are created to model groundwater flow patterns using computer of 2930 Intel Aspire computer, Intel® Core ™ Duo processor T6400 (2.0 Ghz, 800 MHz FSB, 2 MB L2 cache), Mobile Intel® Graphics Media Accelerator 4500 MHD, 1 GB DDR2, 12.1" WXGA Acer Crystal Brite™ LCD, 250 GB HDD, DVD-Super Multi DL, and 802.11 a/b/g/draft-N WLAN.

The data required to perform the simulation model of groundwater flow in Karst region are the physical properties of the medium Maros karst region that includes, coefficient of permeability, porosity and density of the medium as shown in Table 1.

This study uses the time $t$ and $t + n$ as the data to simulate fluid flow, so that:

1. Data on the physical properties of the medium are constant, the porosity ($\phi$), soil particle density ($\rho$), and the permeability of the medium ($\kappa$) as mentioned in the table 1.

2. Simulation with Lattice-Gas is the model used to facilitate molecular movement known as the Lattice-Gas Automata. Model Lattice-Gas Boltzmann consists of particles that move from one cell to another in a triangular lattice. Using the lattice equilateral triangle of each particle can have six possible directions speed. Each particle has a mass unit and one unit of speed. The total momentum of the collision is zero and the particle rotated 60 degrees. It is because the particles in the Gas-Lattice Boltzmann are described as a Boolean variable $n_i(x,t)$, with the direction determined by the speed unit (Wolf-Gladrow, 2000):

$$c_i = \left[ \cos\left(\frac{2i}{6}\right), \ \sin\left(\frac{2i}{6}\right) \right] \quad \text{with } i = 1, 2, \dots 6 \tag{1}$$

Thus, the particle moves by following a mathematical equation, and it is written as follows:

$$n_i(x+c_i, t+1) = n_i(x,t) + \Delta_i(n_i(x,t)) \tag{2}$$

$n_i(x+c_i, t+1)$ is a Boolean. If there are particles then it value is 1, and 0 if no moving particles from the collision position $x$ to position $x + c_i$ at time $t + 1$.

For notation $n_i(x,t)$, it expressed a particle moving to the collision position $x$ of the direction $c_i$ at time $t$. As for $\Delta_i(n_i(x,t))$ is the collision operator that describes changes in the value $n_i(x, t)$. These collisions can be valuable operator 0, +1, or -1. If there is no change in the number of particles in the direction $i$ caused by collision events, i.e. the number of particles before and after the collision in the direction $i$ are the same, then the value of $n_i(x,t) + \Delta_i(n_i(x,t))$ is 0.

Table 1. Physical Medium Price of Maros karst region

| Dimensions | Unit | Soil Types | |
|---|---|---|---|
| | | Clay | Chalk |
| Specific Graffiti (Gs) | - | 2.80 | 2.61 |
| Wet Density, $\rho_{wet} = (4)/(5)$ | Gr/Cm$^3$ | 1.05 | 1.74 |
| Porosity ($\phi$) | % | 77.30 (High) | 45.17 (High) |
| Permeability Coefficient ($\kappa$) (Constant Head Permeameter) | cm/det | 8.14E-03 (Low) | 7.33E-08 (Low) |
| Permeability Coefficient ($\kappa$) (Falling Head Permeameter) | cm/det | 1.35E-03 (Low) | 7.33E-08 (Low) |
| Degree of Saturation (Sr) | % | 53.48 | 68.16 |

The price in Table 1, further processed to obtain the movement of ground water medium in Maros Karst region, the algorithm as follows

1) Input parameters of time duration flow by N units of time,
2) Create a medium with properties of the medium as mentioned in Table 1,
3) placing the initial position of particles in a triangular lattice, the total momentum of the collision is zero and the particles allows rotated 60 degrees (Rothman, 1988),
4) Particles in the Boltzmann lattice gas can be described as a Boolean variable $n_i$ (x,t), the direction is determined by the speed unit of

$$c_i = \left[ \cos\left(\frac{2i}{6}\right),\ \sin\left(\frac{2i}{6}\right) \right]$$

with $i = 1, 2, \ldots 6$.

5) Particle motion, every one unit of time, the particles move in the direction of the lattice unit speed.
6) Tracking was done at each lattice configuration to determine whether there was the possibility of change in the direction of the particles caused by the collision with a wall or occurs whether or not the configuration, so that : a) if there was a change in the direction of the particle or configuration changes, then the transformation was done on the basis of the data, and b ) if there is no change in the direction of the particle or configuration alterations, then the program move straight to step 6
7) Repeat steps (1) to (5) then it should be performed until the time parameter achieve N as unit time given input.
8) Display the results
9) Finish.

The simulation began with the movement of fluid due to the difference in pressure on both the left and right sides. Fluid moves until it reaches equilibrium. Equilibrium occurs if the time is still moving, but the flow has shown a similar pattern.

That is, the time is growing but the flow pattern remains the same. Therefore, the simulation stops if the flow pattern does not change anymore.

In this study, extraction/pumping out of water are done. The movement of groundwater in medium Karst areas is very different from medium of non-Karst. The heterogeneous of medium characteristic effect on fluid motion. Continuously of
5   fluid motion obtain changes of pressure gradient will cause the process of kartisification becomes slow. Karstification will determine the characteristic of constantly changing of medium in proportion to time. If the pattern of flow shown in this simulation is not changed, then karstification started, thus permeable of medium karst becomes impermeable, so that the medium becomes a water reservoir that holds water in a relatively long time. This process will lead to the karst region has uvala, polje and karst characteristics as water tendon is different from other non-karst medium. In addition, the selection of a
10   barrier in the form of plate adapted to the characteristic of medium karst is a vertical barrier wall. Dimensions of plate depending on the size of the Lattice Boltzmann that allows particles to move with 6 possible directions as equation are as follows (1).

For barriers of circle adapted to the characteristic of medium karst that has undergone karstification, karstification occur**S** if the medium was rigid and stiff, because dissolution causes deformation. This deformation caused barrier fluid movement
15   turns into a circle particles with dimensions larger than the dimensions of the particles that started from 16, 32, etc.

Tabel 2. Material Type in Mine Used Area (Karst)

| No | Color | Resistivity (Ωm) | Material Type |
|----|-------|------------------|---------------|
| 1. |  | 0,169 | Groundwater, sand, clay, and limestone |
| 2. |  | 0,366 | |
| 3. |  | 0,792 | |
| 4. |  | 1,710 | |
| 5. |  | 3,780 | |
| 6. |  | 7,990 | |

(Source: Arsyad, Sulistiawaty & Tiwow, 2016)

20   Tabel 3. Material Type in Mining Area (Karst)

| Resistivity | Mineral |
|-------------|---------|
| 0,0363 Ωm – 0,0784 Ωm | Chalcopyrite
Galena
Pyrite
Pyrrhotite |

| Resistivity | Mineral |
|---|---|
| | Magnesite |
| | Cassiterite |
| 0,0784 Ωm – 0,169 Ωm | Hematite |
| | Pyrite |
| | Magnitite |
| | Chalcopyrite |
| | Cassiterite |
| 0,169 Ωm – 0,366 Ωm | Hematite |
| | Magnitite |
| | Cassiterite |
| 0,366 Ωm – 0,792 Ωm | Magnitite |
| | Cassiterite |
| 0,792 Ωm – 1,71 Ωm | Lempung |
| | Magnitite |
| | Cassiterite |
| 1,71 Ωm – 3,77 Ωm | Lempung |
| | Magnitite |
| | Cassiterite |
| 3,77 Ωm – 7,91 Ωm | Lempung |
| | Magnitite |
| | Cassiterite |

(Source: Arsyad, Sulistiawaty & Tiwow, 2016)

**3 Result and Discussion**

Simulation models of groundwater flow in Maros karst region were divided into two major parts of the groundwater flow model of 16 particle size with a barrier in the form of plates and circles, as well as ground water flow model of 32 particle size with a barrier in the form of plates and circle, at a different time variation to reach steady state. The results are shown in Figure 1 and Figure 2.

[Figure]

[Figure]

[Figure]

**Figure 1: Groundwater flow model of 16 particle size with barrier plate and circle for the duration of time (a) 2.254.016.412 unit of time (b) 10.825.515.496 unit of time (c) 1.088.451.742 unit of time (d) 10.704.906.537 unit of time**

From the simulation results, Figure 2 and Figure 3, it is seen that the motion of the fluid has a tendency specific motion. In general, fluid motion every through the barrier, the speed was getting smaller and tend to be irregular. This irregularity was caused by the viscosity of the fluid with a barrier. Laminar fluid moves in the middle and back unstable movement when approaching the barrier. Fluid particles upon reaching the barrier were moving downwards and so on following the current line. The greater the time duration, the more different fluid motion appears. For the state of the Karst region, this means that the heterogeneous karst medium and highly unstable greatly affect the movement of the particles. Fluid flow at a greater duration of time showing the pattern of fluid flow was enlarged over the barrier in the bottom, as shown in succession Figure 2a through 2c.

[Figure]

[Figure]

**Figure 2. Groundwater flow model of 32 particle size with barrier plate and circle for the duration of time. (a) 2254.016.412 unit of time (b) 11.601.807.596 unit of time (c) 2.254.016.412 unit of time (d) 11.844.106.125 unit of time**

The simulation results indicated that the karstification process takes place in a relatively long time with adequate rainfall. Therefore, the situation is very dependent on the state of the micro-climate of the region, especially the state of rainfall.

Rainfall is one of elements of the climate which is associated with the other elements of the climate, such as the ambient temperature and humidity. Study conducted by (Ahmad and Hashim, 2010) (Sanusi et al., 2015) showed that a change in one element of the climate will affect the other climatic elements, such as the air temperature increases followed by increased rainfall. Furthermore, the high temperatures in the area where the water is in short supply, will lead to an increase in air humidity. If this situation occurs in Pangkep Maros karst region, it will certainly affect the karstification in the region. In addition, this process is also affected by karst topography are available and endogenous processes in karst itself. The result was a cycle of water in karst areas is not followed by the water cycle.

Water collected in the karst hills on epikarst zone is slowly through the slits vadose, fracture, and then fill the underground stream that continues to evolve into an underground river. Therefore, springs or underground rivers in Maros Karst Region will have a time delay after rain events for a few moments with the chemical quality of water is relatively good.

**4 Conclusion**

Mineral resource modeling of groundwater in karst mountain area of Maros-Pangkep had been conducted using the Boltzmann lattice gas automata. The result was a simulation of the hydrodynamic flow patterns by viewing them as lattice gas. The simulation results showed that the greater the time duration, the more different fluid motion pattern. For the state of the karst region, this means that the heterogeneous medium karst is very unstable and affects the movement of the particles. In addition, karstification process takes place in a relatively long time with adequate rainfall. This process is not only dependent on karst topography that is available, but also on endogenous processes in karst itself.

**References**

Aharonof, E. and Rothman, D. H.: Non-Newtonian flow (through porous media): A lattice-Boltzmann method, Geophys. Res. Lett., 20(8), 679–682, 1993.

Ahmad, S. and Hashim, N. M.: Perubahan iklim mikro di Malaysia, Penerbit Fak. Sains Sos. dan Kemanus. Univ. Kebangs. Malaysia, Bangi, 2010.

Arsyad, M.: Simulasi Aliran Fluida Non Newtonian Melalui Media Pori dengan Metode Gas-Kisi Boltzmann, ITB Bandung., 1999.

Arsyad, M.: Survey Potensi Hidrologi di Kawasan Karst Maros-Pangkep, Report of Research Collaboration between Universitas Negeri Makassar and Kementerian Lingkungan Hidup Asisten Deputi Urusan Wilayah Sumapapua. Universitas Negeri Makassar, Makassar, Indonesia., 2002.

Arsyad, M.: Eksplorasi, Eksploitasi, dan Pemodelan Sumber Daya Mineral Air Bawah Tanah di Kawasan Gunung Karts Maros-Pangkep dengan Metode Automata Gas Kisi Boltzmann, Report of 1st years Competitive Grant Research of UNM Makassar. Universitas Negeri Makassar, Makassar, Indonesia., 2009.

Arsyad, M.: Eksplorasi, Eksploitasi, dan Pemodelan Sumber Daya Mineral Air Bawah Tanah di Kawasan Gunung Karts Maros-Pangkep dengan Metode Automata Gas Kisi Boltzmann, Report of 2nd years Competitive Grant Research of UNM Makassar. Universitas Negeri Makassar, Makassar, Indonesia., 2010a.

Arsyad, M.: Estimasi Keberadaan Air Tanah Di Kawasan Karst Maros Sulawesi Selatan, in Prosiding Seminar Nasional Fisika 2010, edited by E. Sustini and A. Singarimbun, Himpunan Fisika Indonesia Cabang Jawa Barat, Bandung., 2010b.

Arsyad, M., Sulistiawaty and Tiwow, V.A. Characteristics Analysis of sediments in the Regions Ex-Mine Karst Maros. Research Report of PNBP FMIPA UNM. Makassar, Indonesia, 2016.

Hughes, W. F., Brighton, J. A. and Winowich, N.: Schaum's Outline of Fluid Dynamics, 3rd ed., McGraw-Hill Education, New York, NY., 1999.

Khotimah, S. N. and Liong, T. H.: Pengembangan metode komputasi dan simulasi, J. KFI, 2002.

Rothman, D. H.: Cellular-automaton fluids: A model for flow in porous media, Geophysics, 53(4), 509–518, 1988.

Rothman, D. H.: Macroscopic laws for immiscible two-phase flow in porous media: Results From numerical experiments, J. Geophys. Res. Solid Earth, 95(B6), 8663–8674, 1990.

Sanusi, W., Jemain, A. A., Zin, W. Z. W. and Zahari, M.: The drought characteristics using the first-order homogeneous Markov chain of monthly rainfall data in peninsular Malaysia, Water Resour. Manag., 29(5), 1523–1539, 2015.

Supriyatno, E.: Penggunaan Model Friyzh-Hasslacher-Pomeau Pada Simulasi Aliran Fluida 2 Dimensi, ITB Bandung., 1998.

Wolf-Gladrow, D. a.: Lattice Gas Cellular Automata and Lattice Boltzmann Models., 2000.